# Survey of Sensitization to Common Fungi in an Allergic Dog Population: The Need for Further Focus on Sensitization and Allergy to Fungi in Veterinary Medicine

**DOI:** 10.3390/jof9111075

**Published:** 2023-11-03

**Authors:** Luís Miguel Lourenço Martins

**Affiliations:** Department of Veterinary Medicine, School of Science and Technology, MED—Mediterranean Institute for Agriculture, Environment and Development & CHANGE—Global Change and Sustainability Institute, Institute for Advanced Studies and Research, University of Évora, Pólo da Mitra, Apartado 94, 7006-554 Évora, Portugal; lmlm@uevora.pt

**Keywords:** dermatomycosis, allergy, *Alternaria*, *Aspergillus*, dermatophytes, fungal allergy, *Malassezia*, *Penicillium*

## Abstract

Most fungal species are commensals and non-pathogenic to plants, humans, or animals. However, several species of the *Alternaria*, *Aspergillus*, *Trichophyton*, and *Microsporum* genera are common causes of disease, even for immunocompetent individuals. Besides mucosal damage, fungi may contribute to a skin barrier impairment, favoring sensitization and allergy development. A total of 68 allergic dogs were selected from a veterinary dermatology and allergy outpatient consultation for conditions related to both *Malassezia* overgrowth and other fungal complications. The allergy diagnosis was made through anamnesis and current clinical criteria, with the involved allergenic species being identified by intradermal tests (IDTs) and allergen-specific immunoglobulin E (sIgE) determination in serum. *Dermatophagoides farinae*, *Dactylis glomerata*, and *Malassezia pachydermatis* showed as the higher sensitization species from house dust mites, grass pollen, and fungi, respectively. Significant correlations at *p* < 0.05 were found between sensitization to *Dactylis glomerata* and *Phleum pratense* grass pollens, *Dermatophagoides farinae* and *Dermatophagoides pteronyssinus*, *Acarus siro*, *Tyrophagus putrescentiae*, and *Lepidoglyphus destructor* dust/storage mites, and between fungi like *Aspergillus* mix and *Penicillium* or *Alternaria alternata*. A significant correlation was also found between sensitization to the *Aspergillus* mix and *D. farinae*, *D. pteronyssinus*, or *A. siro*. Rather severe dermatitis was observed when a positive IDT to *Malassezia pachydermatis* was found, regardless of the detection of circulating sIgE, allowing us to consider the usefulness of both the IDT and the sIgE for a systematic diagnosis of allergy to fungi.

## 1. Introduction

Most fungi have evolved for approximately 1.5 billion years [1] and belong to saprophytic species that are non-pathogenic to plants, humans, or animals. Nevertheless, a small part may, in fact, become pathogenic, either by producing toxins or infecting or sensitizing other living beings, leading to subsequent allergy development [2]. Species from the Fungi kingdom are ubiquitous, like some from the *Alternaria*, *Aspergillus*, *Fusarium*, *Mucor* [2], *Trichophyton*, and *Microsporum* [3] genera and may be a possible cause of disease to humans and animals, mostly associated with immune-compromised conditions [2].

As organic matter decomposers, fungi secrete enzymes into the surrounding environment, digesting molecules from other organisms like carbohydrates, proteins, and lipids and then absorbing the resulting nutrients (e.g., carbohydrate metabolites) in a heterotrophic way. An eventual enrichment of the surrounding environment in fungal metabolic leftovers occurs due to the airborne spread of spores, hyphae, and their fragments [4].

Different species of fungi have been recognized as sensitizers and are possibly allergenic. Besides respiratory infections [5], several species of the *Aspergillus* genus have been implicated in bronchopulmonary allergies, sinusitis, and IgE-mediated asthma or hypersensitivity pneumonitis [6,7]. Sensitization to *Alternaria* has also been reported, mostly in warm climates, to be associated with type I hypersensitivity in both indoor and outdoor environments [8,9]. Species of the *Fusarium* genus, common contaminants of cereals, may lead to an immune system impairment and sensitization with allergy. Fungi-related respiratory diseases are frequently associated with either atopy [8] or immune impairment [2]. Allergic alveolitis, atopic conjunctivitis, bronchial asthma, and rhinitis are among the most common manifestations [10].

Allergens from *Fusarium* are known to cross-react with each other [11] and sometimes also with allergens from other species [12]. *Curvularia*, a genus with over 40 mostly saprophytic species, may lead to sensitization, causing mainly respiratory symptoms in humans [13]. It also cross-reacts with *Alternaria alternata* and *Epicoccum nigrum* [14,15]. *Curvularia* infection and allergy have also been reported in dogs [16,17]. *Cladosporium* is another ubiquitous genus with reported infection of humans and dogs [18], horses [19], and cats [20,21], as well as allergies in humans [2,22] and dogs [23]. *Mucor* and *Rhizopus*, two genera from the *Mucorales* group, have also been associated with allergy in humans [8] and animals [17,24,25]. Besides *Alternaria*, *Aspergillus*, *Candida,* or *Epicoccum*, *Cladosporium*, another mostly outdoor genus belonging to the *Ascomycota* phylum, also comprises relevant allergy-causing species with several identified allergens [26].

Dermatophytosis, a condition mainly caused by fungi from the *Microsporum*, *Trichophyton*, and *Epidermophyton* genera, is a common condition in immunocompetent human and animal individuals [27,28] (Figure 1). With several allergens already identified from *Trichophyton*, sensitization and allergy to dermatophytes is a well-known condition. Those allergens may either trigger immediate or delayed hypersensitivity. IgE-mediated asthma has also been reported in fungi-sensitized human patients, while delayed responses seem to provide some protection. Allergens from *Trichophyton* may play a simultaneous role in fungal pathogenesis and allergenicity, as suggested by their amino-acid sequence. This dual condition may be associated with particular T-cell epitopes, which may play a key role in new peptide vaccine development, with better efficiency regarding *Trichophyton* infection and allergy [29].

*Candida* is a genus with over 200 species, most of them commensals in human and animal microbiota and only facultatively pathogenic. A total of 15 species have been isolated from human and animal infections, affecting several organs besides mucosa or skin [30]. *Candida* is not a frequent cause of infections in animals but may occur associated with atopy [31,32].

Sensitization to fungi reaches roughly 5% of the global human population. However, this rate is higher in the atopic population. Exposure to fungal allergens may vary according to the environment, and contact occurs with intact spores, mycelia, or their fragments. Germinating spores present a higher allergen variety. Thus, the environment in which individuals live will play a crucial role in their contact with fungal particles, with fungal structure-derived particles becoming aerosolized in concentrations of 300 to 500 times greater than the original spores [33]. While certainly high, contact with fungal particles may result in underestimated subclinical sensitization. Moreover, contact with primary sensitization to a small number of fungal species may result in sensitization to a wide variety of species, as sensitization to fungi is considered highly cross-reactive. When a group of 6565 human individuals presenting with sIgE to fungi in at least one test was evaluated with a larger set of fungal species, 1208 showed positive to all [34]. Fungal proteins, sharing homologous structures and similar functions, have shown marked cross-reactivity [35,36].

Fungi commonly comprise a high concentration of airborne allergens, but increased exposure to indoor microbial diversity may play a protective role for atopy [8]. However, studies on the *Alternaria* genus, one of the most associated with allergy, have shown that even a low level of exposure to fungi for long periods is not necessarily associated with sensitization, except in atopic individuals.

Recurrent airway obstruction (RAO), a common condition in equines, has been associated with frequent exposure to moldy hay, but only non-IgE-mediated mechanisms have been implicated in the pathogenesis. However, sensitization to fungi with clinical deterioration in moldy environments and by challenging with mold extracts was observed. Higher scores in basophil histamine-releasing tests, upon stimulation with fungal allergens in horses with RAO, were also observed when compared to healthy individuals [37]. Regarding the immune response to *Aspergillus fumigatus*, sIgE and IgG were found in bronchoalveolar lavage (BAL) fluid from RAO-affected horses following in vitro provocation with fungal extracts [38]. Despite showing no difference in sIgE to fungal extracts between healthy and affected horses, sIgE to fungal allergens, such as Alt a 1 and Asp f 7, 8, and 9, were mainly detected in BAL and in serum from RAO-affected patients. Relevant differences in terms of sIgE levels for Asp f7 were also observed between healthy and RAO-affected individuals [39,40].

Despite the lack of identification of the major allergens implicated in sensitization, mold allergomes to horses have already started to be disclosed. Sensitization to fungi in dogs and cats also requires further studies, especially regarding the identification of the implicated allergomes [41].

Skin testing is currently performed with whole-allergen extracts, which may vary in allergen content, compromising inter-brand and even inter-batch reproducibility. After a clinical diagnosis of atopy, the IDT (Figure 2) stands as the veterinary allergy first-line diagnosis method for implicated species. Currently, most suppliers provide allergen extracts in well-defined concentrations for most groups of allergenic species [42], although standardization still needs improvement to ensure better reproducibility [43].

*Malassezia*, a relevant genus from a lipophilic group of yeasts [44], is a common commensal of the dog and cat skin and mucosa, with *Malassezia pachydermatis* as the most reported species out of 18 [45].

Despite the common commensal equilibrium, different factors such as innate/adaptive immune impairment and secreted virulence factors may allow the *Malassezia* population to overgrow, leading to severe dermatitis [45], often observed in allergic dogs, with or without sensitization to *Malassezia* [42] (Figure 3).

The inflammatory skin conditions, mostly associated with an impaired epidermal surface, sebum production, or excessive moisture, may favor *Malassezia* overgrowth in a faulty context of the complex cutaneous equilibrium. In fact, besides keratinization disorders, several endocrinopathies, metabolic diseases, neoplasia, and atopic dermatitis, food hypersensitivity or flea allergy frequently trigger a set of conditions commonly associated with *Malassezia pachydermatis* overgrowth, leading to dermatitis [46]. *Malassezia* overgrowth in dogs usually presents as ceruminous pruritic external otitis or as intertriginous dermatitis. The skin becomes erythematous, evolving to kerato-sebaceous skin thickening [47].

Despite its clinical relevance, not many studies have been published in veterinary medicine regarding *Malassezia*-derived animal conditions [43]. In 2021, Di Tomaso et al. [48] reported the results of sera evaluation from 45 dogs, in which 11 showed positive for *Malassezia*. Another study evaluating both IDTs and serum sIgE to *Malassezia* showed 24% positivity in IDTs despite no positive sera [49]. In 2009, Furiani et al. [50] had already reported 35.5% positive IDTs.

In a previous study with 111 mainly atopic dogs, including 33 from predisposed breeds, mostly with seborrheic disruptive skin barrier, 49.6% showed *Malassezia* overgrowth with dermatitis [51]. In yet another study with 84 mainly atopic dogs, including 31 from predisposed breeds, 58.3% presented with *Malassezia* overgrowth and dermatitis [52].

Regarding *Malassezia*, dog breeds, such as the American Cocker Spaniel, Australian Silky Terrier, Basset Hound, Boxer, Dachshund, English Poodle, Setter, Shih Tzu, and West Highland White Terrier have been appointed as being at higher risk of overgrowth, leading to dermatitis. For cats, the Devon Rex and Sphynx are the two breeds recognized as more predisposed to *Malassezia* overgrowth [53]. This condition has, however, been mostly diagnosed in individuals suffering from visceral paraneoplastic syndromes [47]. Zoonotic risks are assumed to be low, especially in immunocompetent individuals [45].

Considering an increase in the number of allergic animals, especially in dogs, where clinical deterioration is associated with mycotic complications like the ones related to *Malassezia* overgrowth and the fact that only a limited number of studies on this subject is available, this research aimed to frame the prevalence of sensitization to the most common fungi in an allergic dog population, by the two main current diagnostic methods, IDTs and sIgE determination in serum.

## 2. Materials and Methods

### 2.1. Dog Population

A total of 68 allergic dogs were selected for conditions related to *Malassezia* overgrowth and/or other fungal complications from the University of Évora Veterinary Hospital (Évora, Portugal) dermatology and allergy outpatient consultation. Primary allergy diagnosis/selection was made through a comprehensive query for anamnestic and clinical criteria, according to Hensel et al. (2015) [43] and Olivry and Mueller (2020) [54]. The owners of all animals presented for consultation at the Veterinary Hospital of the University of Évora were informed and consented to the collection and storage of data, including for research purposes.

### 2.2. Methods

Dogs were subjected to further diagnostic evaluation through IDT by inoculation of 50 μL of commercial allergen extracts (Diater and Nextmune, Madrid, Spain) from a set of relevant allergenic species, such as *Dactylis glomerata* and *Phleum pratense* grass pollens, *Dermatophagoides farinae*, *Dermatophagoides pteronyssinus*, *Acarus siro*, *Tyrophagus putrescentiae*, and *Lepidoglyphus destructor* dust/storage mites, and *Alternaria alternata*, the *Aspergillus* mix, and *Malassezia pachydermatis* fungi. Positive (0.01% histamine phosphate solution) and negative (physiological saline solution) commercial controls were also administered (Diater and Nextmune, Madrid, Spain). Dermal wheal and flare reactions were evaluated after 15 min. Reactions were considered positive when the resulting wheals were at least equal to or higher than halfway between the negative and the positive control reaction and then scored increasingly from 0 (negative) to 4 (maximum positive) [43]. All dogs were sedated with commercial medetomidine (Orion Pharma, Espoo, Finland) (roughly 0.03 mg/kg body weight, administered subcutaneously), and their sedation was reversed after the procedure, with the correspondent dose of atipamezole (Orion Pharma, Espoo, Finland) by intramuscular injection. The allergen-sIgE to an environmental panel of allergenic species, including the ones used for IDT, were determined in macELISA (LETI Animal Health Laboratories, Barcelona, Spain), and the results were expressed in ELISA Absorbance Units (EAU). The results above the 150 EAU threshold were considered positive. A total of 30 dogs simultaneously underwent IDT and sIgE determination; 21 of them underwent only sIgE, and 17 underwent only IDT. Differences between intradermal wheal scores and between sIgE EAU were compared using Pearson’s correlation coefficient (www.socscistatistics.com and SPSS, Chicago, IL, USA). Statistical significance was set at *p* < 0.05.

## 3. Results

### 3.1. General Observations

The sIgE and IDT scores varied according to the individual sensitization patterns and skin reactivity, as well as according to each extract potency. sIgE and IDT positive rates are shown in Table 1. The sensitization rate, defined as positive sIgE, showed the following order of magnitude: *D. farinae* > *T. putrescentiae* > *A. siro* > *D. pteronyssinus* > *D. glomerata* > *P. pratense* > *L. desctructor* > *M. pachydermatis* > *Aspergillus* mix. All dog patients with positive IDT to fungal species also showed at least a positive skin test for dust/storage mites or grass pollens. Only three patients with circulating sIgE to a fungal species (*M. pachydermatis*, in case) did not present with sIgE to dust/storage mites or grass pollens. No circulating sIgE was found to *Alternaria alternata* in this population. Regarding IDTs, the decreasing positive rate was as follows: *D. farinae* and *L. destructor* > *A. siro* > *T. putrescentiae* > *D. pteronyssinus* > *A. alternata* > *Aspergilus mix* > *M. pachydermatis* > *D. glomerata*. *Penicillium* was not assessed by IDTs.

House dust and storage mites were clearly shown to be the most allergenic species for this population, followed by the two grass pollen species regarding circulating sIgE, besides the higher rate of positive IDTs to the fungal species. Three dogs with positive IDTs to a grass pollen mix did not present with sIgE to *Dactylis glomerata* or *Phleum pratense*, while nine dogs with sIgE to these species did not simultaneously reveal positive IDTs.

The highest rate of positive sIgE to fungi was found in *M. pachydermatis* (10 out of 51), followed by *Aspergillus* mix (6 out of 51) and *Penicillium* mix (4 out of 51). None of the dogs presented with circulating sIgE to *A. alternata*, despite 9 out of 47 having shown positive IDT, which is a rate close to that observed regarding other fungal species. None of the dogs revealed simultaneous sensitization to the four fungal species. The highest positivity to fungi was found in *M. pachydermatis*. Of the 10 dogs with sIgE and 7 with positive IDT, only 1 was positive in both tests. None of the tested dogs showed simultaneous sIgE and positive IDTs to the *Aspergillus* mix. *Malassezia pachydermatis* was found to be the only sensitizing fungal species for nine dogs, while the *Aspergillus* mix was found in five, and *Alternaria alternata* was found in four. None of the dogs revealed *Penicillium* as a single fungal sensitizer. A total of 23 dogs revealed reactivity to at least one fungal species/group of species, and 14, most of the fungi-reactive patients, revealed reactive to at least two or three species/group of species.

### 3.2. Observed Associations

Regarding the sensitization to house dust mites and storage mites, a significant correlation was found between sIgE to *D. farinae* and *D. pteronyssinus* (r = 0.587; *p* < 0.00001), *A. siro* (r = 0.958; 0.00001), *T. putrescentiae* (r = 0.967; *p* < 0.00001) and *L. destructor* (r = 0.343; *p* = 0.01). In terms of intradermal reactivity, a significant correlation was also found between *D. farinae* and *D. pteronyssinus* (r = 0.384; *p* = 0.007), *A. siro* (r = 0.446; *p* = 0.001), and *L. destructor* (r = 0.336; *p* = 0.02).

Regarding grass pollen, a significant correlation was found between *D. glomerata* and *P. pratense*, both in terms of sIgE (r = 0.885; *p* < 0.00001) or intradermal reactivity (r = 0.31; *p* = 0.027).

For simultaneous sensitization to fungi and house dust or storage mites, a significant correlation was found between the sIgE to *Aspergillus* mix and to *D. farinae* (r = 0.283; *p* = 0.046) or *A. siro* (r = 0.288; *p* = 0.042). Sensitization between the *Aspergillus* mix and the *Penicillium* mix was also correlated (r = 0.356; *p* = 0.011), as well as positive IDTs to *Alternaria alternata* and *Aspergillus mix* (r = 0.599; *p* < 0.00001). Correlation between IDTs to the *Aspergillus* mix and *D. pteronyssinus* was also observed (0.494; *p* = 0.0002).

## 4. Discussion

As ubiquitous forms of life, fungi and their particles are common components of the environment, in contact with mucosal and cutaneous barriers. The depths of penetration will depend on the quality of those barriers. Skin barrier dysfunction is associated with the deep penetration of allergens, which is commonly observed in allergic dogs [55], promoting sensitization [56].

House dust and storage mites are among the most allergenic species for dogs, as was also observed in this study population, with high sensitization and IDT scores. Pollinosis was not found clinically relevant for most patients despite their sensitization to *D. glomerata* and *P. pratense*. In fact, many of these dogs presented sensitization to those pollens without suffering from pollinosis.

Regarding fungal species, sensitization was found to be low, despite an expressive rate of positive IDT, associated with a compatible clinical frame, with consistent improvement following directed antifungal treatment and environmental sanitation measures.

Associations found regarding either sIgE or IDT between dust mites and storage mites [57,58,59] and grass pollen species [60,61,62] were already expected due to cross-reaction. Interestingly, the highest significant correlation (r = 0.967; *p* < 0.00001) observed in terms of sIgE was found between *D. farinae* and *T. putrescentiae,* as suggested by ELISA inhibition studies recently performed by Song et al. (2022) [59]. Sometimes, those underlying sensitizations lead to allergy, and sometimes, it does not [42,63,64]. Only three dogs sensitized to grass pollens did not present with sIgE both to *D. glomerata* and *P. pratense*, while nine were found without positive IDTs for both, which may reveal a higher sensitization rate without allergy. Pollinosis-associated atopic dermatitis would demand not only circulating sIgE but also the presence of sIgE on skin mast cells, leading to positive IDTs. A similar condition was found regarding *M. pachydermtis*, where ten dogs presented with sIgE, but only seven showed positive IDT. On the other hand, as observed by Han et al., 2020 [49] regarding *M. pachydermatis*, in this study, no sIgE to *Alternaria alternata* was detected in serum, despite a significant number of positive IDTs, which may be due to different levels of preservation of IgE epitopes in the extracts used for those different methods. In addition to an individual predisposition, the associations found between sIgE or IDT to dust and storage mites, and those to fungi (e.g., *D. farinae*, *A. siro*, and *D. pteronyssinus*, regarding the *Aspergillus* mix) may also be related to (i) living conditions, favoring an environmental prevalence of both groups of allergenic species, and (ii) an increased skin barrier impairment, promoting the sensitization to fungi, in dogs allergic to both dust and storage mites. Other associations, like the simultaneous sensitization to more than one fungal species, were also observed. Cross-sensitization between fungal species [11,12,14,15,65], as well as between fungi and food [66], has also been reported. However, some of the positive correlations found, such as the one between *D. pteronyssinus*, a less sensitized dust mite for dogs, and the *Aspergillus* mix, for instance, may decay with sample enlargement.

In this study, the highest sensitization rate to fungi was observed regarding *M. pachydermatis*, with 16 positive cases (sIgE and IDT). *Malassezia pachydermatis* was even the single sensitizing fungal species for nine of the patients, which could be explained by the fact that *Malassezia* overgrowth is much more frequent in atopic dog populations and may result in a higher sensitization rate. In addition to the presence of sIgE in *M. pachydermatis*, more severe inflammatory skin conditions were found when positive IDT was observed. Moreover, as with other allergen species, and probably even more, due to the proteolytic capacity inherent to mold extracts, with consequent self-disruption, specific diagnosis of allergy to fungi should also undergo both IDT and sIgE, filling the gaps from each method [43,67].

## 5. Conclusions

Considering the observed results, showing the relevance of sensitization and allergy to fungi in dogs, extensive future studies are needed, aiming to overcome several current limitations associated with (i) different extract origins as well as different methodologies (e.g., none of the dogs presented with circulating sIgE to *A. alternata*, despite 9 out of 47 having shown positive IDTs, and none of the tested dogs having presented sIgE and positive IDTs to the *Aspergillus* mix, simultaneously); (ii) the usefulness of a healthy non-allergic population, allowing the identification of possible non-specific reactions, and (iii) the need for a larger sample, providing a more accurate set of correlation results.

## Figures and Tables

**Figure 1 jof-09-01075-f001:**
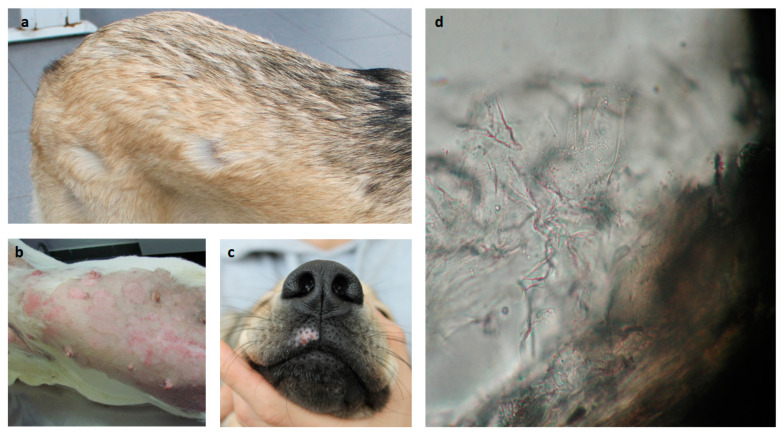
Common dermatomycosis lesion observations in dogs. (**a**) Spots of alopecia; (**b**) pruritic inflammatory spots of alopecia (in previously clipped area); (**c**) pruritic alopecic and depigmented spot; and (**d**) surrounding hair debris with refringent spores (400×).

**Figure 2 jof-09-01075-f002:**
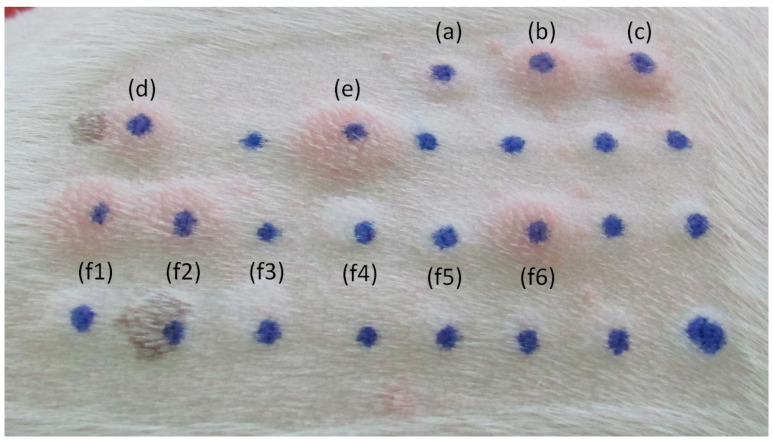
Common IDT reactions in a 0–4 score range in dogs. (**a**) Negative control; (**b**) positive control; (**c**) *Phleum pratense*; (**d**) *Dactylis glomerata*; (**e**) *Malassezia pachydermatis*; and (**f1**–**f6**) house dust and storage mites.

**Figure 3 jof-09-01075-f003:**
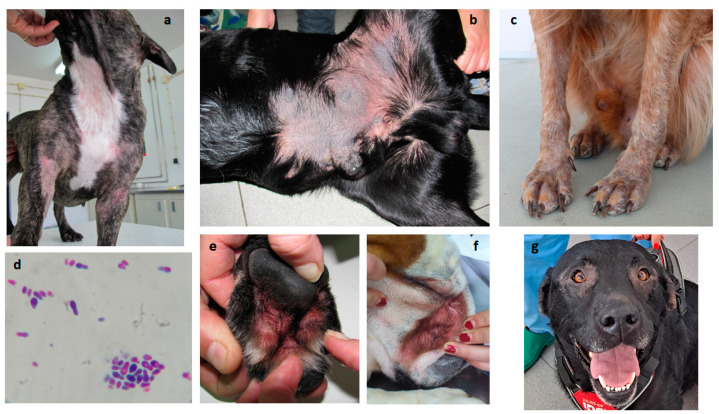
Pruritic dermatitis in allergic dogs presenting *Malassezia* overgrowth. (**a**) Common areas of alopecic acute allergic dermatitis; (**b**) hyperkeratosis and lichenification in chronic dermatitis; (**c**) alopecia and early hyperkeratosis; (**d**) *Malassezia* overgrowth in skin cytology (400×); (**e**) chronic interdigital dermatitis; (**f**) facial intertriginous dermatitis; and (**g**) periocular alopecia in allergic dermatitis.

**Table 1 jof-09-01075-t001:** Number of positive responses to IDT and sIgE testing for examined allergenic species.

	*D. glomerata*	*P. pratense*	*D. farinae*	*D. pteronyssinus*	*A. Siro*	*T. putrescentiae*	*L. destructor*	*A. Alternata*	*Aspergilus mix*	*M. pachidermatis*	*Penicilium mix*
sIgE	18	17	40	27	36	39	12	0	6	10	4
IDT	5	7	24	11	20	12	24	9	8	7	-

*Penicillium* mix was not assessed in IDT.

## Data Availability

The data are not publicly available due to ethical/deontological secrecy.

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
