# Peer review of "Survey of Sensitization to Common Fungi in an Allergic Dog Population: The Need for Further Focus on Sensitization and Allergy to Fungi in Veterinary Medicine"

_jof, 2023, doi:10.3390/jof9111075_

Round 1

Reviewer 1 Report

Comments and Suggestions for Authors

English needs deep revision as the grammar is inappropriate in most paragraphs and often it is difficult to understand the ideas of the authors.

I consider title as irrelevant as, obviously, article deals with (as far as I could understand from the description) various allergic sensitization in dogs, sensitized to fungi. So, the title should be changes accordingly. E.g. “Survey of allergic co-sensitization (patterns, peculiarities, specifics) in fungi-sensitized dogs”

No clear conclusions provided

A lot of Italic-style representations are missed in species names, check, please

One more general recommendation pertains sensitization and allergy. Please, while revising English, care about the meaning of sentences too. I mean, keep in mind, please, that sensitization is not always develops to clinical allergy and allergy (if you consider clinical manifestation while writing “allergy”) is the stage, which follows allergic sensitization.

Abstract:

Despite the fact that you claim in the title that the presented article deals with sensitization to common fungi in an allergic dogs, in Abstract you are writing that “Dermatophagoides farinae, Dactylis glomerata and Malassezia pachydermatis showed as the higher sensitization species from each group.” This is disappointing as Dermatophagoides farinae and Dactylis glomerata are house dust mite and grass accordingly. This is one but not only the reason why I think that the title should be adjusted with the Manuscript content.

What is more, you wrote: “Significant correlations at p<0.05 were found between Dactylis glomerata and Phleum pratense grass pollens, Dermatophagoides farinae and Dermatophagoides pteronyssinus…” However, obviously, correlation was found between levels, frequency etc, of sensitization to these species. This sentence and similar, which can be found throughout the text should be changed accordingly.

Lines 21-22: you wrote: “allowing us to perceive (or suppose? This word looks better) the usefulness of both sIgE and IDT for allergy diagnosis to fungi “in dogs” should be added here as you study dogs in this case.

Introduction:

This chapter discusses just sensitization to fungi, and co-sensitization to fungi and other allergens is not mentioned here. This information should be added as Article deals with study of polysensitized dogs.

Lines 29-30: “or sensitizing with subsequent allergy” should be changed to: “or sensitizing other living beings with subsequent clinical allergy development”.

Lines 34: “Fungi” should be replaced by “fungi”/

Lines 45-63 text should be revised much as English is poor here and many expressions are incorrect or poor understandable (E.g. Delayed responses seem to confer protection, while immediate not…” – protection of who? What protection? – not clear or “which may play a key role in new peptide vaccines” - should be: “which may play a key role in new peptide vaccines development” etc., );

Also, in line 53 you mentioned Mucorales group for Mucor and Rhizopus, but higher taxa for other fungi are nor given (Ascomycota for Alternaria, Cladosporium, Basidiomycota for Fusarium, Malassezia etc.)

Lines 79-92: information should be clarified as it is nor always understandable if it pertains to dogs or humans;

Line 103: your mentioned patients. It is better to clarify what patients do you mena – horses, dogs or humans, not clear much.

Figure 2. It is not clear neither form the figure caption nor from the text, what animals were tested, whom wheels are presented. Despite the fact that generally article deals with dogs, other animals are mentioned here (horses, cats), so Figure caption should be clarified.

Methods:

Again: not clear of you indicate in the Article title that just fugal sensitization is considered why in Methods your describe other than fungal diagnostic tests. Method, results and article title should be harmonized, as it has been pointed out earlier.

Line 179: “Sedation was reversed post-testing” – this expression is not clear and grammatically incorrect,

The same for lines 183-184: “Thirty dogs were simultaneously submitted to IDT and sIgE determination”, should be: “Thirty dogs simultaneously underwent IDT and sIgE determination etc.” or another verb can be used.

Also, it is not clear if some IgE-sensitization threshold was applied. For example, sensitization in humans is considered positive to some allergens, if sIgE level is 0,35 kU/L for ISAAC and 0,31 kU/L for ALEX test, for example, What was the threshold in this case? Was it implicated at all?

Results:

Table 1.

The title can be changed as follows: “Number of positive responses to IDT and sIgE testing for examined allergenic species” or so.

Lines 201-203: “Only 3 dogs sensitized to grass pollens did not present with sIgE to both D. glomerata and Phleum pratense, while 9 were without positive IDT to both.” – the meaning is not clear, should be reconsidered during the English revision.

Lines 204-206: While describing fungal sensitization patterns in dogs, mention, please, if they were co-sensitized with other allerges or just sensitized to fungi. It is not clear at all from this text.

Lines 224-226 – What correlation is mentioned – between IDT or sIgE levels? Also, correlation is observed between sensitivity (!) to some allergens but not between allergens by themselves (this should be clarified in the entire paragraph, especially in lines 227-232).

Discussion:

Needs revision in terms of providing better, clearer association between own Author’s results and literature data.

Lines 239-240 – which population is mentioned? Probably, fungi-sensitized dogs? Or population investigated in the presented study? Clarify, please.

Line 247: you wrote: “Associations found regarding sIgE or IDT, between dust and storage mites” – but association is found between either IDT- or sIgE-determined sensitization to these allergens, not between sIgE or IDT or between dust and sporage mites by themselves, but between levels, fact of this sensitization presence etc. Change accordingly.

Lines 253-257 – the explanation is not clear enough, should be revised.

Comments on the Quality of English Language

English grammar is poor, the entire text needs revision

Author Response

English needs deep revision as the grammar is inappropriate in most paragraphs and often it is difficult to understand the ideas of the authors.

The author agrees to prepare a better English writing.

I consider title as irrelevant as, obviously, article deals with (as far as I could understand from the description) various allergic sensitization in dogs, sensitized to fungi. So, the title should be changes accordingly. E.g. “Survey of allergic co-sensitization (patterns, peculiarities, specifics) in fungi-sensitized dogs”

Disagree as the focus is current sensitization and allergy to fungi, besides the co-sensitization frame, and not the global sensitization and allergy panorama in a fungi-sensitized dog population. The less addressed sensitization and allergy to fungi in dogs is precisely the focus of this brief report research manuscript. However, a rather consensual title will be proposed.

No clear conclusions provided

It may be improved aiming a better highlight of the main goals, identified through the found results. However, facing the relatively limited knowledge on this filed of allergy in dogs, only clear result-derived conclusions were made, drawing attention to what the author thought relevant to further clinical knowledge development.

A lot of Italic-style representations are missed in species names, check, please

The author thanks for calling the attention.

One more general recommendation pertains sensitization and allergy. Please, while revising English, care about the meaning of sentences too. I mean, keep in mind, please, that sensitization is not always develops to clinical allergy and allergy (if you consider clinical manifestation while writing “allergy”) is the stage, which follows allergic sensitization.

The author agrees and is aware of those concepts and will revise accordingly.

Abstract:

Despite the fact that you claim in the title that the presented article deals with sensitization to common fungi in an allergic dogs, in Abstract you are writing that “Dermatophagoides farinae, Dactylis glomerata and Malassezia pachydermatis showed as the higher sensitization species from each group.” This is disappointing as Dermatophagoides farinae and Dactylis glomerata are house dust mite and grass accordingly. This is one but not only the reason why I think that the title should be adjusted with the Manuscript content.

The author thanks for the observation and agrees to revise the manuscript as the statement may me have resulted misunderstood. In fact, when referring to “each group” the author effectively meant house-dust mites, grass, and fungi, respectively.

What is more, you wrote: “Significant correlations at p<0.05 were found between Dactylis glomerata and Phleum pratense grass pollens, Dermatophagoides farinae and Dermatophagoides pteronyssinus…” However, obviously, correlation was found between levels, frequency etc, of sensitization to these species. This sentence and similar, which can be found throughout the text should be changed accordingly.

The author thanks and agrees to include those data (e.g. frequency and levels of sIgE, standard deviation) for the different allergen species mentioned.

Lines 21-22: you wrote: “allowing us to perceive (or suppose? This word looks better) the usefulness of both sIgE and IDT for allergy diagnosis to fungi “in dogs” should be added here as you study dogs in this case.

The author understands the remark but proposes “…allowing us to consider…” instead.

Introduction:

This chapter discusses just sensitization to fungi, and co-sensitization to fungi and other allergens is not mentioned here. This information should be added as Article deals with study of polysensitized dogs.

The author understands the concern. However, the aim of the article as brief report is just sensitization and allergy to fungi, obviously framing the sensitization and allergy to a few other very common sensitizing species. A more complete approach would be more suitable for a “Article” option (≥4000 words).

Lines 29-30: “or sensitizing with subsequent allergy” should be changed to: “or sensitizing other living beings with subsequent clinical allergy development”.

The author makes no objection and will change.

Lines 34: “Fungi” should be replaced by “fungi”/

It was a mistake and will be corrected accordingly.

Lines 45-63 text should be revised much as English is poor here and many expressions are incorrect or poor understandable (E.g. Delayed responses seem to confer protection, while immediate not…” – protection of who? What protection? – not clear or “which may play a key role in new peptide vaccines” - should be: “which may play a key role in new peptide vaccines development” etc., );

The author thanks and agrees to rewrite.

Also, in line 53 you mentioned Mucorales group for Mucor and Rhizopus, but higher taxa for other fungi are nor given (Ascomycota for Alternaria, Cladosporium, Basidiomycota for Fusarium, Malassezia etc.)

The author agrees to complete the information.

Lines 79-92: information should be clarified as it is nor always understandable if it pertains to dogs or humans;

The author agrees to clarify the information.

Line 103: your mentioned patients. It is better to clarify what patients do you mena – horses, dogs or humans, not clear much.

The author agrees to clarify the information.

Figure 2. It is not clear neither form the figure caption nor from the text, what animals were tested, whom wheels are presented. Despite the fact that generally article deals with dogs, other animals are mentioned here (horses, cats), so Figure caption should be clarified.

The author thanks for the especially relevant remark and agrees to complete the information/caption.

Methods:

Again: not clear of you indicate in the Article title that just fugal sensitization is considered why in Methods your describe other than fungal diagnostic tests. Method, results and article title should be harmonized, as it has been pointed out earlier.

The author asks for clarification regarding the mentioned procedures, besides the related to the essential diagnostic methods as IDT and macELISA. Sorry as the focus of the remark was not understood.

Line 179: “Sedation was reversed post-testing” – this expression is not clear and grammatically incorrect,

The author proposes to change to “…sedation was reversed after the procedure…”.

The same for lines 183-184: “Thirty dogs were simultaneously submitted to IDT and sIgE determination”, should be: “Thirty dogs simultaneously underwent IDT and sIgE determination etc.” or another verb can be used.

The author thanks for the remark and will correct the sentence.

Also, it is not clear if some IgE-sensitization threshold was applied. For example, sensitization in humans is considered positive to some allergens, if sIgE level is 0,35 kU/L for ISAAC and 0,31 kU/L for ALEX test, for example, What was the threshold in this case? Was it implicated at all?

Yes, it was. A threshold of 150 EAU was applied. The author will include that information.

Results:

Table 1.

The title can be changed as follows: “Number of positive responses to IDT and sIgE testing for examined allergenic species” or so.

The author agrees.

Lines 201-203: “Only 3 dogs sensitized to grass pollens did not present with sIgE to both D. glomerata and Phleum pratense, while 9 were without positive IDT to both.” – the meaning is not clear, should be reconsidered during the English revision.

The author agrees.

Lines 204-206: While describing fungal sensitization patterns in dogs, mention, please, if they were co-sensitized with other allerges or just sensitized to fungi. It is not clear at all from this text.

Despite the referred simultaneous sensitization to different fungal species: i) sensitization between Aspergillus mix and Penicillium showed positive correlation (r=0.356; p=0.011) as well as positive IDT to Alternaria alternata and Aspergillus mix (r=0.599; p<0.00001); ii) sensitization between Aspergillus mix and house dust or storage mites also significantly correlated – with D. farinae (r=0.283; p=0.046) and with A. siro (r=0.288; p=0.042 with). Correlation between Aspergillus mix and D. pteronyssinus was also observed (0.494; p=0.0002).

The author agrees to include more data regarding the further sensitization frame of those patients.

Lines 224-226 – What correlation is mentioned – between IDT or sIgE levels? Also, correlation is observed between sensitivity (!) to some allergens but not between allergens by themselves (this should be clarified in the entire paragraph, especially in lines 227-232).

In that part of the manuscript, the author found his sentence “Interestingly, the highest significant correlation (r=0.967; p<0.00001) observed in sIgE was found between D. farinae and T. putrescentiae as suggested by ELISA inhibition studies recently performed by Song et. al. (2022) [58].” that refers to sIgE.

The author feels somehow embarrassed but does not find the part “Also, correlation is observed between sensitivity (!) to some allergens but not between allergens by themselves (this should be clarified in the entire paragraph, especially in lines 227-232).” in the text. Is it possible to further define this point?

Discussion:

Needs revision in terms of providing better, clearer association between own Author’s results and literature data.

The author agrees to try a better integration of those informations.

Lines 239-240 – which population is mentioned? Probably, fungi-sensitized dogs? Or population investigated in the presented study? Clarify, please.

When the author mentioned “Despite sensitization to the different fungi species tested, the highest rate was observed to M. pachidermatis, with 16 positive cases (sIgE and IDT) and being the only sensitizing species in 9 individuals across this group of dogs, which may be explained by the fact that it was an atopic population, where Malassezia overgrowth is highly frequent.” he is referring to the studied population. However, the author agrees to try to turn the sentence more assertive.

Line 247: you wrote: “Associations found regarding sIgE or IDT, between dust and storage mites” – but association is found between either IDT- or sIgE-determined sensitization to these allergens, not between sIgE or IDT or between dust and sporage mites by themselves, but between levels, fact of this sensitization presence etc. Change accordingly.

When the author wrote (Lines 224-226) “Associations found regarding sIgE or IDT, between dust and storage mites [56,57,58] and between grass pollen species [59,60,61] were expected as genetic predisposition to develop sensitization as well as cross-reaction occur.” he was referring to: i) associations between sIgE to different dust and storage mites, and between IDT to those; also to ii) associations between IDT to different grass pollen species, and between IDT to those. The author will rewrite the sentence to clarify.

Lines 253-257 – the explanation is not clear enough, should be revised.

The author does not have discussion text beyond L-252. Is it possible to transcribe the text in consideration?

Comments on the Quality of English Language

English grammar is poor, the entire text needs revision

The author agrees to prepare a better English writing.

Reviewer 2 Report

Comments and Suggestions for Authors

The selection of the study population appears to be only based on Malassezia overgrowth without a clinical assessment for atopic dermatitis. This needs to be clarified. How many of the dogs would be considered to have atopic dermatitis on clinical assessment? How was dermatitis assessed? It is no clear what reference 53 has to do with the assessment of atopic dermatitis.

The study of Farver et al. (2005) (DOI: 10.1111/j.1365-3164.2005.00463.x), which is not cited here, reports that Malassezia extract had an irritant effect when used at 1000 PNU mL. Was attention paid to this?

The study does not have a sample of responses of non-atopic heathy dogs to help sort out specific and non-specific reactions.

The abstract and the concluding paragraph in the paper emphasise the correlation between the severity of dermatitis and the responses to Malassezia extract. No data on this were presented and no method for assessing the severity of dermatitis was given.

The source of the allergen extracts used for ID tests needs to be given especially given the possibility of irritant reactions for some.

The cut off for a positive ID test was for a reaction to be more than half the size of the difference between the control and the histamine positive control wheal. This is rather unusual and make positivity dependent on the size of the histamine response. It is not explained why an allergic responses half the size of the histamine injection would be irrelevant. A threshold for allergen concentration would be better as would the wheal size. 

The paper does not give any assessment of the degree of the allergic reactions or the amount of IgE. Correlations with weak reactions are not very helpful and have led to misleading conclusions.

The IgE assessments were measured in absorbance units. This has little value without some quantitative measure as clearly show by the many misconceptions produced by 20th century studies of human allergy.

Comments on the Quality of English Language

OK

Author Response

The selection of the study population appears to be only based on Malassezia overgrowth without a clinical assessment for atopic dermatitis. This needs to be clarified. How many of the dogs would be considered to have atopic dermatitis on clinical assessment? How was dermatitis assessed? It is no clear what reference 53 has to do with the assessment of atopic dermatitis.

In L-145-151 (Materials and Methods - Dog population) the author refers to the primary diagnosis of allergy (mainly atopy and food allergy) using references 42 and 53 (others more basic are also referred throughout the manuscript). The author also refers to the selection of 68 dogs from among the whole allergic population, for reasons of “…Malassezia overgrowth and/or other fungal complications.”. So, the 68 dogs were primary diagnosed allergic and with further Malassezia overgrowth and/or other fungal complications. The author considers reference 53 as a current scientific state of the art regarding diagnosis of food allergy, which is relevant to also identify food allergy conditions.

The study of Farver et al. (2005) (DOI: 10.1111/j.1365-3164.2005.00463.x), which is not cited here, reports that Malassezia extract had an irritant effect when used at 1000 PNU mL. Was attention paid to this?

Yes. The concentration of the commercial extract is truly below.

The study does not have a sample of responses of non-atopic heathy dogs to help sort out specific and non-specific reactions.

It is true. However, in clinical practice it is known that there is frequent to find sensitization with positive IDT without allergy to the implicated species (of course it may be also associated to cross-reaction to species not present in the patient living environment). The author is aware of that limitation and even tried, in the past, IDT with commercial extracts and with low doses (1 to 15 μg) of molecular allergens in perfectly healthy adult dogs, obtaining truly positive reactions. In fact, diagnosis of allergy, either atopy or food allergy, is a clinical issue. Then, IDT and sIgE determination should be valued only in presence of allergy (and if allergen-specific immunotherapy is considered as therapeutic option).

The abstract and the concluding paragraph in the paper emphasise the correlation between the severity of dermatitis and the responses to Malassezia extract. No data on this were presented and no method for assessing the severity of dermatitis was given.

It is true. The author mentions that sensitization to Malassezia is frequently related to severe dermatitis, when comparing to other cases where Malassezia overgrowth is observed without evidence of sensitization. That has been clinically evident. However, this fungal allergy approach was made based on retrospective data and CADESI-4 (the more objective assessment method in use in our Hospital) was not extensively applied apart of prospective studies. Despite not many cases identified so far with positive IDT/sIgE to Malassezia (respectively 7 and 9) severity of dermatitis stood out clearly.

The source of the allergen extracts used for ID tests needs to be given especially given the possibility of irritant reactions for some.

Thank you for the remark. In L-159 it was stated (Diater and Nextmune, Madrid, Spain) but by lapse it did not result clear regarding the allergen extracts. It will be added.

The cut off for a positive ID test was for a reaction to be more than half the size of the difference between the control and the histamine positive control wheal. This is rather unusual and make positivity dependent on the size of the histamine response. It is not explained why an allergic responses half the size of the histamine injection would be irrelevant. A threshold for allergen concentration would be better as would the wheal size. 

The author followed a published scientific reference for objective purposes. However, the author is aware of reviewer’s concern, which is also his. In several situations, results below that minimum are registered as suspicious, for further clinical consideration. Unfortunately, for this manuscript the author had to follow a given scientific threshold, not forgetting that biological activity differences even between batches from the same provider are also not negligible.

The paper does not give any assessment of the degree of the allergic reactions or the amount of IgE. Correlations with weak reactions are not very helpful and have led to misleading conclusions.

The assessment of severity was not frequently available, but clear clinical allergic conditions were found in each and every patient. All of the positive IDT were clearly considered as positive (please see previous response) and positive sIgE were only considered for Laboratory results above the threshold (>150 EAU). The author is also aware that a relation between clinical severity and level of IDT response or sIgE is inconsistent. 

The IgE assessments were measured in absorbance units. This has little value without some quantitative measure as clearly show by the many misconceptions produced by 20th century studies of human allergy.

It is, in fact. That is an issue to which veterinarian laboratorial research has to pay attention in the near future.

Comments on the Quality of English Language

English quality will be addressed.

Reviewer 3 Report

Comments and Suggestions for Authors

Dear author,

I read your manuscript concerning a survey of sensitization to common fungi in an allergic dog population. The paper is about an interesting argument and the manuscript could be improved. Some points should be addressed.

1)     The title doesn’t reflect the whole manuscript; I suggest improving it and reporting more details. Refer to APA style.

2)     The abstract is not fluent and lacks information that could help the reader understand the clinical setting and the results. I suggest to re-organize it, maybe in the first step using a structured abstract. In the second step, you should write an unstructured one.

3)     At the end of the discussion, you should report the aim of your study.

4)     In the material and method section, there is a lack of aims, goals, and software used in the statistical analysis. Refer to other publications in the field:

-        Dauvillier J, Ter Woort F, van Erck-Westergren E. Fungi in respiratory samples of horses with inflammatory airway disease. J Vet Intern Med. 2019;33(2):968-975. doi:10.1111/jvim.15397

-        Martins L. M. L. (2022). Allergy to Fungi in Veterinary Medicine: Alternaria, Dermatophytes and Malassezia Pay the Bill!. Journal of fungi (Basel, Switzerland)8(3), 235. https://doi.org/10.3390/jof8030235

5)     Dermatophytosis is not only considered zoonosis, correct.

6)     Lines 34,66, 137, 151, 241, and 242 correct

7)     Do you not agree to any approval given by a local ethical committee or from the approval given by the owners of the dogs?

8)     Check reference style.

9)     Reference number after Author et al., not at the end of the sentence.

10) The methodology and the results are clear and well-explained. However, the introduction and discussion are poorly organised and difficult to read. Moreover, the study's limitations are missing, and it would be interesting to report a few lines on the epidemiological and clinical implications of the study. 

Comments on the Quality of English Language

Minor editing of English language required

Author Response

Dear author,

I read your manuscript concerning a survey of sensitization to common fungi in an allergic dog population. The paper is about an interesting argument and the manuscript could be improved. Some points should be addressed.

1)     The title doesn’t reflect the whole manuscript; I suggest improving it and reporting more details. Refer to APA style.

The author thanks the remark and will address that point.

2)     The abstract is not fluent and lacks information that could help the reader understand the clinical setting and the results. I suggest to re-organize it, maybe in the first step using a structured abstract. In the second step, you should write an unstructured one.

The author thanks the remark and will also address that point.

3)     At the end of the discussion, you should report the aim of your study.

The author thanks the remark and will also address that point, especially at the end of the Introduction section.

4)     In the material and method section, there is a lack of aims, goals, and software used in the statistical analysis. Refer to other publications in the field:

-        Dauvillier J, Ter Woort F, van Erck-Westergren E. Fungi in respiratory samples of horses with inflammatory airway disease. J Vet Intern Med. 2019;33(2):968-975. doi:10.1111/jvim.15397

-        Martins L. M. L. (2022). Allergy to Fungi in Veterinary Medicine: Alternaria, Dermatophytes and Malassezia Pay the Bill!. Journal of fungi (Basel, Switzerland)8(3), 235. https://doi.org/10.3390/jof8030235

The author thanks the remark. Please see the previous response.

5)     Dermatophytosis is not only considered zoonosis, correct.

Yes, as it has been recently addressed regarding veterinary medicine by Martins (2022) https://doi.org/10.3390/jof8030235

6)     Lines 34,66, 137, 151, 241, and 242 correct

L- 34: The author referred “…possible cause of disease in people and animals…”. Please clarify.

L-66: The author referred “…only facultatively pathogenic…”. Please clarify as Candida is not a Dermatophyte.

L-137, 151, 241-242: The author cannot identify the text. Please clarify as the Line numbering may differ between reviewer and author.

7)     Do you not agree to any approval given by a local ethical committee or from the approval given by the owners of the dogs?

The university of the author demands ethical committee approval for studies concerning prospective non common clinical approaches. All owners agree with the use of results from their pets for scientific and teaching purposes (following strict legal data protection).

8)     Check reference style.

The author will address this point.

9)     Reference number after Author et al., not at the end of the sentence.

 Thank you for the remark. The author had that doubt. It will be corrected.

10) The methodology and the results are clear and well-explained. However, the introduction and discussion are poorly organised and difficult to read. Moreover, the study's limitations are missing, and it would be interesting to report a few lines on the epidemiological and clinical implications of the study. 

The author will re-address Introduction and Discussion organization to turn it easier and more understandable. Study limitations will be addressed. One or two paragraphs will be added regarding epidemiology and aimed clinical implications.  

Comments on the Quality of English Language

Minor editing of English language required

English editing will be also addressed.

Round 2

Reviewer 1 Report

Comments and Suggestions for Authors

Thank you for your work on the article.
My new suggestions are as follows:

English still needs revision. This pertains newly-added sentences too.

It is still illogical why the title of the Article reflects just sensitization to fungi if the Abstract, Methods and Results include description of dog sensitization to other allergens as well.

This is my comment, which you haven't considered: "Again: not clear much – if you indicate in the Article title that just fugal sensitization is considered, so why in Methods you describe diagnostic tests that let determine dog sensitivity to other than fungal allergens. Method, results and article title’ should be harmonized, as it has been pointed out earlier." Hope, now it can be understood better. In my primary comments I misspelled "if" at the beginning of this note writing "of" instead and it changed the sense.

Line 210-213 of the revised text include mention of patients again. These patients were dogs? Or humans? It should be clarified or another way of description should be used. I would suggest to write “dog patients”

Lines 214-217. You wrote: Regarding IDT, the decreasing positive rate was: D. farinae and L. destructor > A. siro > T. putrescentiae > D. pteronyssinus > A. alternata > Aspergilus mix > M. pachydermatis > D. glomerata > Penicillium. Penicillium was not assessed by IDT. 

If Penicillium (sensitization! not Penicillium by itself) was not assessed, so how it can be rated here?

“House dust mite” and “storage mite” are separate words, but sometimes you are writing “house dust-mite”, “storage-mite”. This should be checked and changed.

Also, your result when IDT to Alternaria was positive and sIgE to this fungi were not determined in any (dog?) patient, may be related to violation of the methodology of IDT performance. This procedure should be done at the absence of any exacerbations of allergy in dogs. So, in case dogs experienced any allergy symptoms in in the period of your intervention, your IDT result can be false positive. How did you prevent this possible false positivity during the study? Have you done so?

Line 281-283 of your text explaining what I am saying now. There you wrote: "in this study no sIgE to Alternaria alternata was detected in serum, despite an expressive number of positive IDT, which may be related to different preservation of IgE epitopes in extracts for different methods." - so, due to exacerbations of allergy and impure application of the method, you could obtain false positive result.

Going a little back. On lines: 256-257, you wrote: Deep penetration will depend on the quality of those barriers... Should be: "The depths of penetration...."

New sentence, added in lines 257-259, A low skin barrier effect is considered associated with deeper allergen penetration, which is commonly seen in allergic dogs, allowing higher penetration with possible sensitization [56]. - also needs revision of English.

Your conclusions also support my idea that the experiment could nor be well-done with considerations of all. So, it is important to explain now well, why your present results can be published and, the last but not least, why it should be Journal of Fungi, as there still a lot of data not pertains fungal sensitisation. And revise English, please.

Comments on the Quality of English Language

English still needs revision including newly-added paragraphs. 

Author Response

Reviewer 1

Thank you for your work on the article.
My new suggestions are as follows:

English still needs revision. This pertains newly-added sentences too.

English was revised, including the newly-added sentences. The author maintains a mild worry regarding this issue as the best English reviewer should be also an expert in the field, hence an author, which is not possible at this stage of the manuscript. Nevertheless, the author foresees that the manuscript is now in an acceptable English. Revised sentences are in red text.

It is still illogical why the title of the Article reflects just sensitization to fungi if the Abstract, Methods and Results include description of dog sensitization to other allergens as well.

This is my comment, which you haven't considered: "Again: not clear much – if you indicate in the Article title that just fugal sensitization is considered, so why in Methods you describe diagnostic tests that let determine dog sensitivity to other than fungal allergens. Method, results and article title’ should be harmonized, as it has been pointed out earlier." Hope, now it can be understood better. In my primary comments I misspelled "if" at the beginning of this note writing "of" instead and it changed the sense.

Dear reviewer, thank you once again for the further review, addressing a few issues that could have been not so well explained before as well as new issues.

Regarding the coherence between the title and the further sections where the study population phenotype is characterized: The manuscript focuses on an allergic fungi-sensitized dog population and related diagnosis, which demands framing regarding, at least, sensitization to the most common allergenic species. In fact, most patients showed polysensitized, needing an appropriated approach, both through the avoidance and allergen-specific immunotherapy. If I am not mistaken, a single focus on sensitization to fungi, despite the global sensitization and allergy frame would not allow achieving the useful conclusions obtained.  The title "Survey of sensitization to common fungi in an allergic dog population: The need for further focus on sensitization and allergy to fungi in veterinary medicine." points to the sensitization to common fungi in an allergic population. However, the author is wide-open minded regarding a title which could better mirror this brief-report manuscript.

Line 210-213 of the revised text include mention of patients again. These patients were dogs? Or humans? It should be clarified or another way of description should be used. I would suggest to write “dog patients”

The author changed to “…dog patients…” clarifying the text. However, the manuscript methods and results only report to dog patients. So, the author hopes it won’t result in possible redundance.

Lines 214-217. You wrote: Regarding IDT, the decreasing positive rate was: D. farinae and L. destructor > A. siro > T. putrescentiae > D. pteronyssinus > A. alternata > Aspergilus mix > M. pachydermatis > D. glomerata > Penicillium. Penicillium was not assessed by IDT. 

If Penicillium (sensitization! not Penicillium by itself) was not assessed, so how it can be rated here?

This was author´s mistake when revising. Thank you. Reference to Penicillium was eliminated there.

“House dust mite” and “storage mite” are separate words, but sometimes you are writing “house dust-mite”, “storage-mite”. This should be checked and changed.

It was checked and corrected.

Also, your result when IDT to Alternaria was positive and sIgE to this fungi were not determined in any (dog?) patient, may be related to violation of the methodology of IDT performance. This procedure should be done at the absence of any exacerbations of allergy in dogs. So, in case dogs experienced any allergy symptoms in in the period of your intervention, your IDT result can be false positive. How did you prevent this possible false positivity during the study? Have you done so?

Thank you for rising this matter, which is of crucial relevance, firstly because those are different methods, with different immune significance to be considered. Besides, all methods present their flaws as the author stated in the discussion and now also conclusions (299-311; now 302-316).

Prevention of either false positive or negative results was extensively taken in consideration, not testing dogs undergoing any exacerbation of the skin condition or after many months of sign remission, respectively. This has been a key-rule for IDT and serum sIgE evaluation of patients since the author started performing allergy diagnosis consultation in 2009. This rule as well as respecting the withdrawal periods for different medications and using the appropriate sedation protocols have been followed as good practices since the first day. The author truly hopes these doubts are just for confirmation as the author feels somehow embarrassed for being questioned. Sorry for the sincerity.

Line 281-283 of your text explaining what I am saying now. There you wrote: "in this study no sIgE to Alternaria alternata was detected in serum, despite an expressive number of positive IDT, which may be related to different preservation of IgE epitopes in extracts for different methods." - so, due to exacerbations of allergy and impure application of the method, you could obtain false positive result.

The author agrees as false positive results (as well as negative) occur. However, once again, good practices were implemented to avoid that as much as possible: no patient presenting with any skin exacerbation process was tested, and IDT was performed according to the good practices by experienced veterinarian.

Another interesting point of view may be associated to the matter of clinically relevant vs not relevant sIgE. However, the author didn’t want to get in this somehow new discussion, which would stand considerably apart of the manuscript content:

Kumagai A, Nara T, Uematsu M, Kakinuma Y, Saito T, Masuda K (2021). Development and characterization of a unique anti-IgE mouse monoclonal antibody cross-reactive between human and canine IgE. Immun Inflamm Dis. 9:1740-1748. DOI: 10.1002/iid3.531

Going a little back. On lines: 256-257, you wrote: Deep penetration will depend on the quality of those barriers... Should be: "The depths of penetration...."

Changed accordingly.

New sentence, added in lines 257-259, A low skin barrier effect is considered associated with deeper allergen penetration, which is commonly seen in allergic dogs, allowing higher penetration with possible sensitization [56]. - also needs revision of English.

Sentence has been revised.

Your conclusions also support my idea that the experiment could nor be well-done with considerations of all. So, it is important to explain now well, why your present results can be published and, the last but not least, why it should be Journal of Fungi, as there still a lot of data not pertains fungal sensitisation. And revise English, please.

The author thinks that the manuscript, focusing on sensitization and allergy to fungi in allergic dogs, is relevant for two main reasons:

  1. More complete diagnosis of the allergic condition in this species, allowing to establish better avoidance and hygiene measures as well as more directed immunotherapy, will allow a more successful therapy.
  2. ii) Man and dog share the same living environment, contacting with the same aeroallergens and present several equivalent clinical signs and pathways, which will allow them to be mutual models for allergy studies in a one health approach to allergy.

There is a considerable lack of information regarding allergy to fungi, especially in veterinary medicine. The retrospective-based observations regarding the need for checking for sensitization and allergy to the most relevant fungal species for dogs (included in this manuscript) by both IDT and serum sIgE, could stand as an added value for the topic in consideration in this special issue (Fungal Allergen and Mold Allergy Diagnosis). This focus follows the rational presented by Martins (2022): https://doi.org/10.3390/jof8030235. If this focus is considered as a wrong rational the author will simply withdraw the manuscript.

Regarding the inclusion of data besides sensitization to fungi, it is because fungi-sensitized allergic dog patients are not just sensitized and allergic to fungi as already referred. Hence, for population clinical and immune framing, the author considers essential to add that complementary information. 

Reviewer 2 Report

Comments and Suggestions for Authors

The answers to questions have clarified the limitations of a study in the clinical environment.

Author Response

Reviewer 2

The answers to questions have clarified the limitations of a study in the clinical environment.

Ok. Thank you.

Reviewer 3 Report

Comments and Suggestions for Authors

Dear Author,

I read the new version of the manuscript. Most of the points have been addressed, but not all. 

1) Format the main text font.

2) It must be inserted in the text as the consent collection takes place and the storage of data.

3) A separate section, "conclusion", should be included

Comments on the Quality of English Language

Dear Editor,

The paper has been improved, and the paper as a brief report is adequate, but more clarification about ethical statements and informed consent should be reported. 

Author Response

Reviewer 3

Comments and Suggestions for Authors

Dear Author,

I read the new version of the manuscript. Most of the points have been addressed, but not all. 

1) Format the main text font.

The author has formatted those tracks but asks for editor’s help regarding this issue as it could have resulted from the layout edition.

2) It must be inserted in the text as the consent collection takes place and the storage of data.

The information was inserted in the text (L-180-182).

3) A separate section, "conclusion", should be included

Comments on the Quality of English Language

A separate section “conclusions” was included, highlighting those remarks.

An additional revision of the English was carried out following the changes to this version and the author hopes this has improved the text.